## Research Article

mental health plans and policies; Europe; public mental health; global mental health policy; policy making

**Corresponding author:**
Petr Winkler;
Email: petrwin@gmail.com

# Implementation of mental health policies and plans across the WHO European region: Barriers and facilitators

Zoe Guerrero[1,2,3] , Anna Kågström[1,2,4] , Hana Tomaskova[1,2], Akmal Aliev[1,2] , Yongjie Yon[5], Ledia Lazeri[5], Cassie Redlich[5] and Petr Winkler[1,2,6,7]

[1]Department of Public Mental Health, National Institute of Mental Health, Czech Republic; [2]WHO Collaborating Center for Public Mental Health Research and Service Development; [3]Department of Psychology, Faculty of Arts, Charles University, Prague, Czech Republic; [4]Department of Global Public Health, Karolinska Institute, Stockholm, Sweden; [5]World Health Organization Regional Office for Europe, Copenhagen, Denmark; [6]Institute of Psychiatry, Psychology, and Neuroscience, King's College London, London, UK and [7]Department of Social Work, Faculty of Arts, Charles University

## Abstract

Mental health policies and plans (MHPPs) are powerful tools developed to facilitate real-world changes in mental-health-related prevention, promotion and treatment. This study examined barriers and facilitators to MHPP implementation across the WHO European region. Key informants from 53 countries were contacted and 25 provided in-depth qualitative interviews on MHPP existence, implementation, and evaluation related barriers and facilitators of implementation. We analyzed data via qualitative framework analysis approach aligned with the WHO Comprehensive Mental Health Action Plan 2013–2030. Reported facilitators included active involvement of key stakeholders, ongoing mental healthcare reform, bottom-up approach to implementation, sufficient funding, favorable political receptivity and strong monitoring. Barriers encompassed insufficient funding, workforce shortages, adequate training in psychiatry, missing or insufficient infrastructure in terms of both physical structures and technology for data collection, low political receptivity, stigma and bureaucratic obstacles. While notable progress has been made in the development of mental health plans in the European region, substantial gaps remain in information systems, research capacity, and systematic evaluation frameworks on mental health and development of appropriate evaluation plans. Strengthening these components is essential to ensure the effective and sustainable implementation of MHPPs throughout the region.

## Impact statement

Research on the implementation of mental health policies and plans (MHPPs) in Europe is scarce. We draw from studies that were carried out at individual country levels or on smaller subregions of Europe. However, these studies present outcomes of specific activities rather than presenting barriers and facilitators to their implementation and evaluation. In the absence of information regarding the progress of MHPP implementation across the European region, there is a need to establish an evidence base in this regard. This study provides insights into the key facilitators and barriers to the progress of implementation and evaluation of MHPPs across the WHO European region and compares them against the WHO Comprehensive Mental Health Action Plan 2020–2030. This allows us to discern which areas or activities have notable gaps, based on the perspective of local key implementers. Effective intersectoral communication, along with transparent and secure funding, has significantly advanced the development and implementation of mental health plans in the European region. Conversely, barriers such as underdeveloped information systems, shortages of human resources, excessive bureaucracy and persistent stigma have hindered progress. To ensure ongoing improvement, continuous evaluation of the implementation process should be integral to mental health plans, with related evidence and experiences shared both nationally and internationally.

## Introduction

Mental health policies and plans (MHPPs) and their implementation are key public health tools that can help improve the wellbeing of individuals and effectively address the ever-increasing global burden of mental disorders. However, there is a lack of evidence and research describing the processes and extent of implementation of MHPPs (Aliev et al., 2023).

Many European countries have faced challenges related to the development and implementation of MHPPs. These, for instance, include low policy responses from key

policymakers and stakeholders, lack of funding, long-stay institutions and their adherence to the institutionalized model of care, lack of social inclusion and low empowerment of people with lived experience of mental health conditions (PWLE) (Knapp et al., 2007).

Examples of bottom-down and top-up MHPP governance structures in Europe come with a range of facilitators and barriers. In the UK, it has been argued that the implementation of mental health policies has been halted due to a lack of mental healthcare services and/or the use of ineffective programs for prioritizing patients to mental health care, which resulted in decreased accessibility and inequitable use of resources (Bindman et al., 1999; Lurie, 2005). In Spain, mental healthcare policy is divided among its 17 autonomous regions, implying a need for each region to establish its own policy and plan. This might have brought about a faster change since regions adapt the plan to their specific needs; however, similar to the UK example above, large disparities seem to exist across different regions, which is further exacerbated by a lack of monitoring and evaluation at the national level (Salvador-Carulla et al., 2010). In addition, increased deinstitutionalization in Spain seems to be associated with an increase in both privatization of services on the one hand and a demand for social services on the other (Desviat, 2011).

Similar to Spain, Sweden chose not to implement its MHPP at the national level and to allow local authorities to adapt or create their own policies and plans, and it was suggested that the absence of a larger national umbrella of MHPPs may have led to difficulties in equal distribution of care (Fjellfeldt, 2020). Finally, reviews from Eastern European countries show that this region faced challenges related to its communist past, during which progress was halted mainly because of ideologies imposed on professional and lay society (Dlouhy, 2014; Winkler et al., 2017). Similar to some other post-communist scenarios, in Poland, one of the key challenges was the introduction of a new health insurance system. Insurance funds, which pay for the majority of services, also became the main developers of mental health plans. This might result in a power imbalance and a lack of willingness to launch a mental healthcare reform (Puzynski and Moskalewicz, 2001. In Albania, the lack of mental health training to primary healthcare workers resulted in unnecessary referrals to specialized care, which has been considered a major challenge underlining the importance of capacity building and human resources (Keste et al., 2006).

As evidenced by the examples above, MHPPs in the European region have faced a vast variety of challenges, and the European region is diverse in the conceptualization, application and implementation of MHPPs. Gaps in the current mental health systems as well as barriers hindering positive changes are considerable. However, the implementation of MHPPs is as a rule not sufficiently evaluated, which leads to opportunity costs associated with lessons that have not been learned (Aliev et al., 2023; Guerrero et al., 2024). We aim to contribute to overcoming this gap by mapping and analyzing major barriers and facilitators related to both the implementation of MHPPs and the evaluation of the implementation process in the European region. We provide an updated overview of where European mental healthcare systems face bottlenecks and where they can learn from each other, as well as recommendations for the further development and implementation of MHPPs.

## Methods

This paper is part of a larger series of papers previously published that detailed the methodology used for data triangulation (Aliev et al., 2023; Guerrero et al., 2024). We contacted 53 key informants (KIs) from each country in the WHO European region to recruit participants for interviews. KIs are members of the Pan European Mental Health Coalition focal points. They were contacted via email. The initial invitation email included questions on the existence of a MHPP and further questions on evaluation and regional plans. A reminder was sent to those who did not respond to the first email. Following their response, KIs were asked to participate in an interview that would expand on the implementation of their MHPPs and related evaluation. Again, a reminder was sent to those who did not respond to this email.

KIs were not provided with the interview questions in advance. They were assured of anonymity as well as of a presentation of data that would not allow for identifying their country. The initial invitations were sent between November 25 and December 15, 2021, and the reminders were sent from January 10 to January 21, 2022. The interviews took place between January 24 and February 18, 2022. This project was approved by the Ethics Committee of the National Institute of Mental Health, Czech Republic (183/21).

Two researchers (AK and ZG) conducted the interviews with KIs, one as a moderator and one as an observer and transcriber. Interview questions focused primarily on assessing the facilitators and barriers to both the implementation of MHPPs and the evaluation of implementation, following one of two versions of the semistructured interview guide. Two versions of interview guides were developed since some countries have the evaluation process for the implementation of MHPPs and some countries do not (see Figures 1 and 2 in the Appendix for a flow diagram of the semistructured interview guides). Interviews were audio-recorded to facilitate verbatim transcription. KI interviews were analyzed using framework analysis (Gale et al., 2013). The analysis followed five steps: (1) familiarization: all transcripts were read by the coders to gain an overall understanding of the data; (2) identifying a thematic framework: two researchers (ZG and HT) independently conducted open coding on a subset of transcripts and collaboratively developed a coding framework that reflected both inductive insights and deductive codes aligned with the interview guide; (3) indexing: this coding framework was then applied systematically across all transcripts using Atlas.ti software, and coding discrepancies were discussed and resolved through consensus; (4) charting: a coding matrix was created with cases (interviewees) as rows and codes/themes as columns, allowing for within- and cross-case analyses; and (5) mapping and interpretation: themes and subthemes were refined through iterative discussion, guided by both the emergent data and the WHO Comprehensive Mental Health Action Plan 2013–2030 objectives (World Health Organization, 2021). This systematic approach enabled triangulation between researchers and ensured transparency and rigor in theme development. The final themes and their frequencies are presented in Table 1 in the Appendix.

Synthesis of all the above data was presented against the objectives of the WHO Comprehensive Mental Health Action Plan 2013–2030: 1. to strengthen effective leadership and governance for mental health; 2. to provide comprehensive, integrated and responsive mental health and social care services in community-based settings; 3. to implement strategies for promotion and prevention in mental health;

**Table 1.** Summary of barriers and facilitators for implementation and evaluation

| Implementation | | Evaluation | |
|---|---|---|---|
| Barriers | Facilitator | Barriers | Facilitators |
| • Bureaucracy<br> ○ In communication with insurance companies<br> ○ In interministerial communication<br>• Lack of funding<br> ○ Divided financing between state and national insurers<br>• COVID–19<br> ○ Delay in implementation plans<br> ○ Funding taken away<br>• Lack of interministerial communication<br>• Lack of receptivity<br> ○ Lack of adherence to MHPPs<br> ○ Stigma toward mental health/mental health care<br>• Lack of infrastructure<br> ○ Adequate buildings for provision of mental health care<br> ○ Lack of human resources for mental health care | • Plan with clear, transparent strategy of implementation of activities<br>• Activities targeted at increasing mental health literacy and decreasing stigma toward mental health<br>• Good level of financing<br>• Communication with local stakeholders<br>• Involvement and cooperation with key stakeholders leading to sense of ownership<br> ○ Creation of plan<br> ○ Implementation of plan<br>• PWLE involvement<br> ○ Support of PWLE organizations<br>• Involvement of key ministries<br> ○ In terms of manpower dedicated to mental health care and MHPP implementation<br> ○ In terms of communication<br>• Political receptivity<br>• Mental healthcare reform<br> ○ Community-based care and facilities<br> ○ Further education of mental healthcare workforce<br> ○ Care guidelines anchored in law<br>• Plan for mental health prevention and promotion and working group to implement such activities<br>• Detailed plan for suicide prevention<br>• Academic cooperation with local research institutions | • Lack of communication<br> ○ No established method of communication regarding evaluation<br>• No monitoring and evaluation plan<br>• Lack of receptivity<br> ○ Lack of demand and interest in evaluation from key stakeholders<br> ○ Lack of evaluation culture<br> ○ Lack of engagement from local stakeholders<br>• Lack of funding<br>• Lack of national data registry<br>• Lack of infrastructure<br> ○ Lack of technology to collect data such as computers<br> ○ Lack of adequately trained staff to collect and analyze data<br>• Lack of evaluation tools or evaluation standards appropriate for the context of country<br>• Lack of time for time-consuming evaluation | • Sharing best practices with local and international partners<br>• Clearly established measurable indicators<br>• Provision of feedback loops<br>• Collaboration with international agencies such as the WHO |

and 4. to strengthen information systems, evidence and research for mental health (World Health Organization, 2021).

## Results

### Participants

Twenty-five KIs from 25 countries participated in the interviews. KIs were primarily persons in charge of the implementation activities of MHPPs, such as coordinators usually based in local ministries of health or national institutes of mental health.

### Objective 1: Effective leadership and governance for mental health

#### Implementation

The progress of implementation of MHPPs varied widely, with almost all countries implementing their mental health plans to some extent. Some countries had a recent mental health strategy that had not been implemented yet (n = 3), and some countries also reported being in the process of creating or revising action plans either due to changes related to the COVID-19 pandemic or due to the end of an evaluation period (n = 6). Only one country reported having a mental health plan, but not implementing it actively.

Six countries reported a bottom-up approach to implementation, where local service providers or regional authorities implement mental health plans. This includes collaborating on the creation of the mental health plan, deciding and allocating funding, and implementing specific activities. One country also mentioned collaboration with NGOs in the implementation of mental health plans. A bottom-up approach to implementation was also considered by some KIs (where such an approach was taken) to be a facilitator to implementation, since it promotes advocacy from local organizations and local creation of mental health plans. The latter, according to KIs, allows local health authorities to provide care that is tailored to their population and results in more effective funding allocation.

> "I think the construction of the health target is that each federal state has their own health target, and they are more or less the same as the nation-wide targets, this has led to the fact that in each federal state there is a mental health target too. The federal states could be seen as facilitators, on the national level it was clear that mental health is an important topic but now it is also clear at the federal level."

KIs underlined that it is important to have a mental health plan and strategy that is transparent and clear and is created via an inclusive and transparent process. Some KIs also highlighted the need for programs to include activities to decrease stigma and increase mental health literacy, as well as for more services geared toward community care, such as mobile teams.

On the other hand, bureaucracy was mentioned as a barrier to the implementation of MHPPs (n = 2). Bureaucracy was mentioned to hinder not only the implementation at the ministerial level but also the communication with national insurance companies with respect to desired changes in funding.

> "A main problem is the very high level of bureaucracy [...], this is why we have now the 5th or 6th reading of the draft and we have to reach a level of consensus which is very hard to reach."

Three countries implemented their mental health plans via larger projects funded from either national resources or larger European Union funding schemes. One country reported that the lead in implementation is the national mental health institute in collaboration with the ministry of health.

Eight countries mentioned adequate levels of funding as facilitators to implementation. However, lack of funding was most frequently described as a key barrier to implementation. Besides describing a general lack of funding for mental health care, KIs also underlined issues in having divided financing between the state and national insurance provider(s), which often results in an inadequate distribution of fundings as national insurers frequently have different key interest areas in comparison to the mental health plans.

> "[…] all services […] are financed by mandatory medical insurance and they finance but the budget is not enough and usually we don't have a good communication between the medical insurance and the ministry at the service level"

Therefore, KIs highlighted that a facilitator needed for future implementation is not only the correct management of funding but also an increase in funding in general. The COVID-19 pandemic has allowed in some countries for an increase in budget as it increased the burden on mental healthcare services; however, in other countries, it has led to budget cuts as other competing priorities required funding in the healthcare system. Some achievements reported by KIs were related to changes in funding structures, including a reform in the financing of mental health care and achieving state-funded sessions with psychiatrists.

> "Of course there are financial difficulties, we have costed the interventions in 2013 but it goes parallel with awareness of policy makers that this is an area that should be in focus."

Regarding collaboration and stakeholder engagement, the most frequently mentioned lead in the implementation of MHPPs was the ministry of health or department of health (n = 22). Some countries also mentioned interministerial collaboration as a form of leadership in the implementation process (n = 11). Most frequently, the ministry of health was joined by the ministry of social affairs or ministry of education to lead the implementation of national MHPPs. Besides interministerial collaborations, some countries implemented their mental health plans through working groups, which would include ministry representatives as well as key stakeholders such as service providers and service users (n = 9).

Most of the KIs reported that communication regarding implementation is happening frequently and in different forms. Formal communication tended to be defined as official communication between ministries and governmental entities, while informal communication was described as meetings with smaller groups of stakeholders working on specific targets of mental health plans. Some countries had conversations only at the formal level, which often means communication between ministries and key stakeholders and some regularity in meetings organized centrally (n = 6).

However, some countries also reported struggles with establishing communication. Most often, communication is hard to establish between local care providers and the larger ministerial and governmental structures (n = 4). Some countries also reported it is challenging to establish interministerial communication for specific goals in the implementation of mental health plans (n = 3). Furthermore, the COVID-19 pandemic has acted as both a facilitator and a barrier to communication; some countries reported increased communication mainly due to online tools, whereas others reported competing priorities specifically on the side of service providers and therefore decreased communication.

Several countries (n = 16) reported the involvement and cooperation of key stakeholders as a facilitator. Some of the key stakeholders mentioned were international institutions such as the WHO, local academic institutions, national insurers, national public health institutes and NGOs. Involvement of key stakeholders can take the form of collaboration or cooperation in the creation of mental health plans or active implementation. First, KIs reported that creating a sense of ownership among stakeholders is one of the main facilitators to implementation. Second, the involvement of service users was reported to facilitate implementation as well as creation of mental health plans and services (in the case of the inclusion of peer workers). Third, the involvement of governmental structures, primarily at ministerial level, was an important facilitator. This includes communication between ministries and local healthcare providers but also, for example, an increase in human resources in charge of the implementation that is available at ministries. Some countries reported the existence of joint ministry of health and social affairs as a key facilitating factor. Finally, the involvement of service providers was reported as a key facilitator; besides collaboration, this includes lobbying or creation of umbrella organizations for service providers.

> "I think that when we manage to put the service users, research and clinical field together, we can create ownership. We invited the stakeholders to give us their advice, it is key for us to engage people from day one."

Collaboration of other parties was also considered a possible future facilitator. This includes the support, creation and engagement of service user movements or strengthening capacity for leadership in implementation. The engagement of interest groups that can advocate for the needs in mental health care can also act as a facilitator.

Most of the KIs reported that the receptivity toward implementation in their countries is positive (n = 14). At the political and ministerial level, KIs reported the need for correct justification in order to reach a good level of receptivity. Some countries reported that local health authorities and service providers often have a good level of receptivity toward implementation. Other countries, on the contrary, reported mixed levels of receptivity at the local level and at the higher political level, suggesting that implementation is actively promoted only at the specific stakeholder level (n = 5).

> "If we talk about the regions then it depends, it is sometimes hard for them to understand the policies and activities of the mental health plan. So for example at the departmental level of the Ministry of Health they know how it is done, but at the regional level where the implementation is done there may be some unwillingness in general, not only towards mental health."

Lack of receptivity was a key barrier to implementation (n = 23). Lack of motivation from key stakeholders toward mental health plan creation and implementation was often a key barrier. This includes a lack of adherence to mental health plans, disrupted communication among key stakeholders or competing interests. A lack of leadership was also considered a key barrier, as well as a lack of key stakeholders overall. Finally, the lack of prioritization of mental health often resulted in funding for mental health being the first to be cut, rather than reducing budgets in other areas such as general health services, thus creating an additional barrier to implementation. Furthermore, stigma and lack of mental health literacy in the general population, as well as in political structures, was also a barrier to implementation. Stigma not only creates negative representation in the local media, but also leads to a lack of discussion around key topics in mental health.

Conversely, political receptivity can act as a facilitator to implementation. Political receptivity includes mental health being placed high on the political agenda and support from key ministries such as the ministry of health. The pandemic has also led to increased

stakeholder receptivity toward implementing mental health plans, largely due to the heightened awareness of the need for mental health care it brought to light.

With regard to involvement of service user movements and people with lived experience, most KIs reported that service users are involved in the creation and implementation of mental health plans in their country (n = 14). Some reported a mixed level of involvement. When service user organizations were included, they participated in the development of mental health plans and, to some extent, in their implementation, such as through membership in working groups or committees. When service users were not involved or involved minimally, it was usually due to a lack of funding or overall lack of service user movements in the country. Finally, the involvement of people with lived experience, or the creation of a mental health national ambassador designation, was considered a key achievement.

*"Another facilitator is to have representatives of service users be present in all the process because they refocus the aim of the reform on what really matters."*

### Evaluation

Leadership and governance in terms of evaluation was led and operationalized by a great variety of key stakeholders. In some of the countries, the ministry of health leads the evaluation and collects the data or requests the data from key stakeholders (n = 8). In other countries, the national institute for mental health or public health collects and evaluates data on the implementation of MHPPs (n = 3). Mainly in countries with a bottom-up approach to MHPP implementation, evaluation is done by local healthcare providers or regional authorities. Some countries reported that they have created a working group with key stakeholders who work on establishing the evaluation (n = 3). Finally, one country has employed an external evaluator to carry out the evaluation.

Most of the KIs from countries with no evaluation highlighted a lack of communication regarding evaluation; they also described that there is no established method or frequency in the communication and cooperation regarding evaluation. The absence of monitoring and evaluation processes in the planning of mental health plans was also considered a barrier to effective assessment.

*"In many cases there is no control or evaluation, sometimes our president or the cabinet can request an evaluation of the implementation. But it can be evaluated once every three years. But no one does it every year."*

In terms of communication regarding evaluation, most of the countries that do have an existing evaluation also created an evaluation report, which was then disseminated (n = 13). Dissemination methods varied; a vast majority of the countries published the evaluation report with no follow-up activity. Some, however, organized a workshop for key stakeholders to implement recommendations, created feedback loops via specific national organizations or published results in local media.

Cooperation in terms of evaluation was considered a key future facilitator. First, the need for the involvement of stakeholders was highlighted; this includes the need for a lead coordinator of evaluation, involvement of case managers, involvement of all stakeholders to increase general receptivity and buy-in toward evaluation. Second, there was a frequent mention of international cooperation and a need for sharing best practices and experiences with evaluation from abroad; this is particularly the case for countries without an existing evaluation system.

*"The indicator framework is an important step forward, but I think a more global perspective would be more beneficial so that we can compare with other countries."*

Receptivity toward evaluation was often mixed; in order to assess further, KIs were asked about the attitudes toward or culture of practicing evidence-based activities. Culture of evaluation or evidence-based culture is sometimes adhered to; however, some KIs underlined that evidence-based culture is understood at a theoretical level but not always adhered to practically. This culture then often relates to the level of receptivity toward evaluation. A lack of receptivity toward evaluation also posed a barrier, as key stakeholders often showed limited demand or interest in undertaking it.

*"It is very interesting, because I asked for information for the national [evaluation] report and I don't have it yet from all of them. The Ministry of Labour and Social Affairs gave me part of the information and then not the rest. The people sometimes don't consider the [evaluation] report as important. They want to help with the implementation but they don't think the [evaluation] report is key."*

### Objective 2: Comprehensive, integrated and responsive mental health and social care services in community-based settings

### Implementation

Reforming the mental healthcare system, or aspects of it, toward community-based settings was seen as a key facilitator to implementation (n = 30). This was often done by laying a basis of destigmatization and creating grounds for mental health awareness activities. Restoration or establishment of facilities such as psychiatric hospitals that provide care that is more human rights based and community centered was considered an achievement. This includes a decrease in involuntary treatment methods and facilities, creation of community care centers, implementation of mobile teams and multidisciplinary teams and involvement of peer workers. Second, the key component of reforms – deinstitutionalization, including changes in education, training and funding (cooperation with national health insurance) as well as legislation changes when it comes to anchoring mental health plans in the legislation – was considered a key facilitator. Anchoring guidelines for mental health care in legislation was also considered a key achievement. Finally, training of healthcare providers in key aspects of mental healthcare reform such as rights-based care was considered a facilitator to implementation. This most often included developing training for mental healthcare professionals but also general practitioners or including peer workers and peer worker training into the system of care.

*"A lot of support was given to peer workers, we have started a project to help get peer workers into all activities and forums. I think it will lead to the possibility of having peer workers in services."*

Perhaps related to human resources, another reported barrier to implementation was missing infrastructure (n = 19). This presented as both material infrastructure, such as a lack of adequate equipment or buildings and a lack of infrastructure at a policy and political level. For example, having a very decentralized system (as in bottom-up approaches) of health care can create issues, especially in combination with a non-binding mental healthcare plan. However, a top-down approach can sometimes also create unrealistic guidelines or policies, which may contribute to a lack of policy infrastructure. Frequent changes in political structure can also bring about barriers to implementation as each minister or

government brings forth different priorities for state expenditure. Finally, KIs also talked about the lack of infrastructure when it comes to the attitudes of mental healthcare professionals, where mental healthcare professionals are accustomed to old models of care focused on highly medicalized mental health care, which is often opposed to the community care model proposed in national mental health plans.

> *"We had some barriers in the attitudes of psychiatrists, who were scared of losing power and closing down beds. It still remains a barrier with other professionals, there are always professionals who oppose because they are afraid to lose money. But of course this is not our idea, we want to put more money into the system […]"*

Lack of human resources was often cited as a barrier to implementation (n = 9). This includes the lack of mental healthcare professionals but also the lack of human resources at the ministerial level. Lack of mental healthcare professionals is often related to low salaries and a lack of interest in mental health care in general; this lack of human resources is especially apparent in rural settings (n = 2).

> *"And then human resources – specialists for example in districts, there is a lack of professionals."*

In terms of evaluation, some KIs also reported a lack of experience and leadership in human resources toward evaluation. KIs underlined that there is a need for culturally appropriate tools and training of human resources to aid implementation, which includes training for local healthcare providers such as general practitioners (n = 11). Further education of healthcare professionals in mental healthcare specializations was also considered an achievement, as well as the establishment of parallel research infrastructure. Such reformation of mental health care has also led to some achievements in patient outcomes, for example reduction in involuntary admissions, increase in availability of care and decreases in suicide rates.

> *"By far the biggest barrier is the availability of workforce, we are also looking into how to use our workforce more effectively. […] Retention is another thing; we do see quite a lot of people migrating into the private sector"*

### Objective 3: Strategies for promotion and prevention in mental health

#### Implementation

Low mental health literacy can act as a barrier by feeding into the negative perception of mental health plans among both the public and political leaders. However, high knowledge regarding topics around mental health, especially among stakeholders at the political level, can also act as a facilitator to the implementation of mental health plans.

> *"The other facilitating factor is the widespread awareness about promotion prevention and service provision in mental health, it is high on the political agenda and all parties see it as an important factor."*

Several countries mentioned activities related to promotion and prevention as a key achievement; such activities include the creation of a prevention and promotion plan (n = 2) or, for example, a working group for prevention and promotion, including interministerial collaboration.

> *"It is the beginning; the Ministry of Agriculture launched their first plan for mental health prevention or wellbeing prevention. For students, the Ministry of Education invested in the trainings; and suicide*

> *prevention of police is a new commitment of this ministry too. It has never happened before."*

In terms of suicide prevention, countries mentioned having created a separate plan for suicide prevention as an achievement and facilitator. Some countries considered the implementation of new services such as suicide helplines a key achievement.

#### Evaluation

Regarding evaluation, one country reported on actively discussing indicators that are applicable for prevention and promotion activities, while other countries already reported changes in the general population, most importantly a decrease in stigma, monitoring of child and adolescent mental health and a general increase in public awareness toward mental health (n = 4).

> *"[…] we are building a knowledge infrastructure, we are looking at what indicators we want to use for mental health prevention and promotion, mental ill health. I expect we could have threshold values soon once we discuss with stakeholders. I haven't seen one unanimous threshold."*

### Objective 4: Information systems, evidence and research for mental health

#### Implementation

Information systems for the evaluation of data or possible resources described by the KIs were predominantly national database registries (n = 3). Some KIs also reported that academic institutions or research institutions may be a valuable source or resource for data collection. Finally, national statistical or health institutes were also considered a large source of data for evaluation.

> *"But we also have a separate agency […], and they are the ones that are doing the statistics and collecting the routine data. We have made sure that mental health is part of the mandated collection of data."*

#### Evaluation

Out of the 25 countries, 14 reported having an evaluation plan for the assessment of the implementation of their MHPPs; a smaller proportion of countries also reported having no regular evaluation. Countries with an existing evaluation usually had a system of set targets or indicators; however, the level of elaboration of such indicators varied according to KIs. The methodologies used in the evaluation of mental health plans varied widely as well. Some countries employed routine monitoring of data, whereas some countries relied on stakeholder reports at the local level. Furthermore some countries employed a mixed method approach in which both qualitative data and quantitative data were utilized. Some countries employed clinical audits or financial audits to assess the quality of care at the local level. Many KIs underlined the need for more funding for evaluation and the establishment of a monitoring system, such as a national registry, to facilitate evaluation. Having an established system of monitoring for mental health and mental healthcare justifies the needs in the mental healthcare sector, which may lead to political receptivity and increased funding.

> *"We are not good at evaluation unfortunately, because we do not have good data, and that is a big barrier. We are striving to get a good monitoring system, but it has not materialized. Lack of data is a big handicap."*

KIs reported that evaluation allows for clear mapping of needs in terms of financing and human resources. Some KIs reported that the evaluation report provided an important feedback loop for

service providers. Finally, one country reported providing support for service providers collecting data for evaluation.

With respect to monitoring, evidence and research, there is a lack of infrastructure for data collection, for example using old computers for data collection. Furthermore, some countries reported a lack of experience and therefore a lack of human resources to carry out research (n = 2). Related to this, KIs also reported that either the data are not collected or the collected data are not used or interpreted adequately. First, there is a lack of appropriate evaluation tools for the context of specific countries. Second, the data collected are not considered accurate, or the indicators are not comprehensive. The latter point is made in relation to the fact that indicators are set to be temporary and cannot keep up with the changes happening in the mental health-care system.

> "Another barrier is that our mental health evaluation is not based on outcomes but on temporary indicators such as how many discharges or acceptances. But it does not tell you much, the main outcomes such as burden of disease, percentage of disability, suicide rates, satisfaction with services, quality of life they are not included in the formal evaluation, and this is a problem."

In countries where evaluation is established, KIs also described some key challenges faced when carrying out evaluations. Many countries reported difficulties in establishing indicators in advance, which often results in the use of simple outcome indicators, such as completion status or numerical targets, without corresponding measures of quality. Related to this, KIs underlined that there are no standards for evaluation quality, which means that data interpretation is also challenging. KIs also reported that it is hard to collect data at the local level, which is often tied to the fact that local healthcare providers may provide inaccurate data. Evaluation is time-consuming, and the continuity of data collection and sustainability of such practice is challenging (n = 17).

Finally, an achievement and facilitator toward evaluation and research according to KIs was the involvement of international NGOs and international agencies such as the WHO in the evaluation process.

## Summary

The barriers and facilitators to both the implementation of MHPPs and the evaluation of the implementation process are summarized in Table 1.

## Discussion

This study provides insights into facilitators and barriers to the implementation of national MHPPs and the evaluation of the implementation process in the WHO European region. Large gaps within the implementation of evaluation and information systems related to evidence and research were identified. KIs provided a variety of examples of facilitators to implementation, such as involvement of key stakeholders, reform of mental health care, bottom-up approach to implementation, appropriate funding, political receptivity, monitoring and how the COVID-19 pandemic highlighted the need for mental health services and mental health impacts.

Despite lasting commonalities across the European region, it seems that barriers often remain unaddressed, and facilitators remain underutilized common knowledge. In 2003, Jenkins highlighted many of the same facilitators and barriers to the implementation of MHPPs in their paper, as we do now, such as communication, destigmatization, capacity building, appropriate methods of monitoring and evaluation (Jenkins, 2003). This indicates the need to shift from mapping barriers and facilitators to improving actionable approaches toward increasing facilitators and dismantling barriers collectively across Europe. Each country needs to evaluate which factors can become facilitators and which may become barriers. To achieve this, creating working partnerships with countries with similar contexts to share practices seems to be a viable solution, as well as partnerships that offer cross-national strengthening and mentorships. It is also important to note that barriers and facilitators permeate through all the WHO Comprehensive Mental Health Plan objectives; they are not siloed despite being presented as such in the current study.

Besides collaboration and sharing of best practices internationally, it is critical to be able to look back on barriers and facilitators to implementation and adapt appropriately matched approaches to MHPP implementation. The usage of processes that have been evaluated and are therefore evidence-based in policymaking should be considered a key stepping stone toward better reflection of barriers and facilitators to MHPP implementation. In order to do this, it is vital for countries and their respective MHPPs to have adequate systems of monitoring and evaluation in place, including human capital and physical infrastructure. This will often mean educating policymakers and key decision makers on the importance of evidence-based practices (Williamson et al., 2015).

### Contextual considerations and areas for further exploration

The findings of this study highlight a complex and multifaceted landscape of barriers and facilitators to mental health policy implementation and evaluation across countries. Many of the challenges identified, such as insufficient funding, lack of infrastructure, bureaucratic hurdles, poor interministerial coordination and limited human resources, underscore a deeper issue: the limited integration of mental health policies into broader general health sector strategies. This lack of alignment at all levels of the service continuum, from community-based and primary care to tertiary services, has practical implications for the effective implementation of mental health services. It impacts critical functions such as workforce training, clinical supervision, availability of medications and inclusion of mental health indicators in health management information systems. Though central to service delivery, this dimension was beyond the scope of the current analysis.

Furthermore, the degree to which mental health policies are integrated into non-health sectors, such as education, employment, social welfare and criminal justice, remains underexamined. These sectors play a significant role in shaping the experiences and outcomes of individuals with mental illness and are also critical sites for prevention and promotion in mental health. Future research would benefit from a more comprehensive examination of these intersectoral linkages to fully understand the systemic conditions necessary for effective mental healthcare reform.

Finally, areas such as referral systems or services supporting deinstitutionalization were not systematically addressed in this study. These service components warrant further exploration, particularly in relation to how structural and political facilitators, such as national-level strategies, legal frameworks or international partnerships, can support their development.

Overall, these contextual considerations point to the importance of framing MHPPs within a broader health system and strengthening intersectoral collaboration. Future research should aim to examine these intersections more explicitly, especially to identify

models of integration and coordination that are feasible and effective across diverse country contexts.

## Limitations

There are several limitations to the current study. First, this study is limited by its reliance on a single KI per country, all interviewed in English, which may have excluded important perspectives and introduced language bias. This also means that KIs who were more comfortable speaking in English were more capable of sharing in-depth information. However, participants were purposely selected to represent high-level multisectoral expertise, enabling a broad overview across the European region while maintaining study feasibility. Second, all the KIs we interviewed were from countries where a mental health plan is being implemented or soon will be. This means their views can be biased as their countries are already implementing a mental health plan, which implies that some level of receptivity and cooperation already exists. Third, the perspectives of KIs could also be biased based on the position they hold within the system of implementation. Future research should focus on approaching a wider range of key stakeholders from each country, bearing in mind language barriers.

## Conclusion

Progress has been made in the development of mental health plans in the European region. However, there are several areas that still remain underdeveloped, such as information systems, evidence and research on mental health. Implementation systems are frequently more developed than evaluation systems, with the latter still facing large barriers. Collaboration and sharing of best practices are needed in order for countries to be able to establish optimal systems for the implementation and evaluation of national mental health plans.

**Open peer review.** To view the open peer review materials for this article, please visit http://doi.org/10.1017/gmh.2025.10070.

**Supplementary material.** The supplementary material for this article can be found at http://doi.org/10.1017/gmh.2025.10070.

**Author contributions.** ZG and AK planned and coordinated the study and led the writing of the manuscript. AA and HT collaborated on the writing of the manuscript. PW coordinated and supervised the writing of the manuscript. YY, LL and CR collaborated on the revision of the manuscript.

**Financial support.** This work was supported by the World Health Organization Regional Office for Europe.

**Competing interest.** The authors declare none.

**Ethics statement.** This project was approved by the Ethics Committee of the National Institute of Mental Health, Czech Republic (183/21).

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
