## [Reviewer Report]

Summary

This qualitative study, despite its methodological limitations, is a significant contribution to the field. It examines the barriers and facilitators to mental health policy implementation across 25 European countries, addressing a research gap that is often overlooked. The study’s potential is recognized, and with the necessary revisions, it can make a valuable impact on mental health policy implementation.

Strengths

The study tackles a relevant topic with compits rehensive geographic coverage and across 25 European countries. Its the WHO Mental Health Action Plan as an organizing framework. It and its us on implementation and evaluation provides valuable insights from key informants directly involved in policy execution.

Major Concerns

Methodological Limitations: The study relies on one key informant per country, which is insufficient for robust qualitative research. Conducting all interviews in English introduces substantial bias, likely excluding important perspectives. Additionally, selecting informants through advocacy networks may have resulted in a biased sample of only those already committed to the cause.

Limited Novel Insights: The authors acknowledge that the findings essentially replicate barriers and facilitators identified by Jenkins (2003), raising questions about the study’s unique contributions. The analysis is predominantly descriptive, lacking in-depth critical analysis.

Rigor Issues: Details regarding data saturation, the interview guides used, and systematic analytical procedures are missing. The absence of triangulation with other data sources weakens the validity of the findings.

Recommendations

The paper, while in need of major revisions, presents a clear path for future research. Strengthening methodological reporting, enhancing analytical depth, and articulating unique contributions are key areas for improvement. The potential for future research to consider mixed-methods approaches, address language barriers, and include multiple informants from each country is promising and should inspire further exploration in this field.

Decision: Major Revision Required

While the topic is important, significant improvements in methodology and analysis are necessary for the paper to meet publication standards.

---

## [Reviewer Report]

This is an interesting paper about an important topic. I think my overall concern is that the KI interview guide is very general (aligned to the 4 overall WHO objectives rather than to more specific policy components), and so I felt missed an opportunity to capture more detail on barriers and facilitators in individual countries. For example, one specific barrier which I have found is very common in countries is the lack of integration of the mental health policy into the general health sector strategy at very level of the service (from community and primary care to tertiary care), which then has an impact on practical issues such as training, supervision, medicine supply and HMIS; and this integration of mental health policy with general health strategy is not explored here. There were no specific questions on primary care, referral pathways, community care of people with severe mental illness, decentralisation of services especially local acute beds, supported housing, rehabilitation facilities, and I think each may have provided a rich source of information about barriers and facilitators across countries with which to assist future implementation. Similarly, one wants to know how far mental health policy is integrated into non-health policies such as social welfare, employment, education and the criminal justice system, as each of these sectors wield a large influence on the trajectory of people with mental illness, as well as being important settings for mental health promotion and prevention.

Results section Objective 1, “Three countries implemented their mental health plan via larger projects..” Were these demonstration projects or systemic changes across whole regions of the country?

“Funding is considered as a key facilitator as well as barrier to implementation” This needs rewording. I think what the authors mean is “ ADEQUATE FUNDING IS A FACILITATOR TO IMPLEMENTATION WHILE LACK OF FUNDING IS A BARRIER” Since this is the gist of the following two sentences in fact the sentence is unnecessary and could be deleted.

The paragraph on collaboration and stakeholder engagement should examine how far there is intersectoral coordination at every level of the health system (National, regional, district and PHC levels) , and what the barriers and facilitators are for this to happen.

In the paragraph starting “Lack of receptivity..”, the phrase “A TENDENCY OF FIRST CUTTING” needs rewording/more explanation. Do you mean that health authorities with inadequate funding are inclined to cut mental health services rather than general health services unless mental health is explicitly prioritised?

Under objective 2 it would be useful to know whether geographic distribution of workforce is important....frequently specialists gravitate to the big cities leaving more remote and rural regions very understaffed. The specialist workforce may also migrate to richer countries. It would be interesting to know how far this is a problem for the countries in eastern Europe .

Objective 3 “Mental health promotion and prevention is both a barrier to progress and a facilitator to advancement” This sentence also needs rewording or omitting.

Objective 4 , para starting “ In countries where evaluation is established...” , do you mean “.......which often results in having NO outcome indicators ...”

---

## [Reviewer Report]

The manuscript aims to assess facilitators and barriers to the implementation in the real world of mental health policies and plans in the European region. The issue is important and the work the authors did to gather the material deserves attention. Nonetheless, I found the paper quite confusing, hard to read, and not very informative.

In particular:

- the Introduction focuses on some countries and not others, without a clear rationale for this choice. Such lack of clarity can be found throughout the entire manuscript.

- A major problem lies in the analysis. The authors used a qualitative framework analysis. It allows for flexibility and use of the data to answer specific objectives, that the authors derived from the WHO Comprehensive Mental Health Action Plan 2013-2023. Nonetheless, like other methods, the coding should lead to subthemes and themes. These are the core results of the study, and they feed the answers to the questions/objectives. We do not know about themes and sub-themes. Indeed, the results are grouped “tautologically” into the four objectives and into two parts (“Implementation” and “Evaluation”) in all of them. In light of this, the use of quotations makes little sense. They are often misleading, and reflects only one of the concepts expressed above.

- The Results are affected by the use of the objectives as if they were themes. The answer to the four objectives is not clear. The section corresponding to Objective 1 is disproportionately long and hard to read.

- Some sentences are hardly understandable. For instance, page 7 lines 3-9, or page 9 lines 10-25, and several others. Sometimes the verb is missing.

- The discussion brings some order to the issues raised in the Results, without any synthesis and in-depth interpretation (which is expected in a qualitative study).

- Minor problems contribute to making the reading quite tiring. For instance, on page 2 line 50 the term “power in-balance” probably means “power imbalance”. There are several mistakes, and terms unusual in scientific papers (like “first cutting”). The English needs to be improved.

Assuming that the analysis was appropriately conducted (but there is no evidence of it in the manuscript), the presentation should be completely rearranged to give the reader a clearer and more informative view of the issue of mental health policies implementation in Europe.

---

## [Editor Report]

Thank you for submitting your manuscript “Implementation of mental health policies and plans across WHO European region: barriers and facilitators” to Cambridge Prisms: Global Mental Health. All reviewers indicate that the study is important. However, all reviewers indicate substantial weaknesses in the methodology. Accordingly, you are required to make substantive revisions, attending to the reviewers' concerns and re-submit.

---

## [Reviewer Report]

This study is a solid and well-executed qualitative research that makes a significant contribution to mental health policy research. While the findings somewhat reaffirm existing knowledge about implementation barriers, the systematic perspective across Europe and alignment with the WHO framework provide valuable insights for policymakers.

The research effectively highlights the important gap between policy development and evaluation systems, which is a crucial finding for the field. Although the paper would benefit from a deeper analysis and more specific recommendations, it represents a valuable addition to the literature on mental health policy implementation in Europe.

---

## [Reviewer Report]

The authors have done their best to address the reviewers' comments. I remain disappointed that the methodology was not more searching.